# "We are pleading for the government to do more": Road user perspectives on the magnitude, contributing factors, and potential solutions to road traffic injuries and deaths in Ghana

**Aldina Mesic**[1]*, **Barclay Stewart**[2,3], **Irene Opoku**[4], **Bradley H. Wagenaar**[1,5], **Bilal Andoh Mohammed**[6], **Sulemana Abdul Matinue**[1], **Manal Jmaileh**[1], **James Damsere-Derry**[2], **Adam Gyedu**[6], **Charles Mock**[1,2,3,5,6], **Angela Kitali**[7], **Daniel Hardy Wuaku**[8], **Martin Owusu Afram**[8], **Caryl Feldacker**[1]

1 Department of Global Health, University of Washington, Seattle, Washington, United States of America,
2 Department of Surgery, University of Washington, Seattle, Washington, United States of America,
3 Harborview Injury Prevention and Research Center, Seattle, Washington, United States of America,
4 Building and Road Research Institute, Kumasi, Ghana, 5 Department of Epidemiology, University of Washington, Seattle, Washington, United States of America, 6 Department of Surgery, School of Medicine and Dentistry, Kwame Nkrumah University of Science and Technology, Kumasi, Ghana, 7 Civil Engineering Program, University of Washington, Tacoma, Washington, United States of America, 8 National Road Safety Authority, Accra, Ghana

* amesic@uw.edu

## Abstract

Road traffic collisions disproportionately impact Ghana and other low- and middle-income countries. This study explored road user perspectives regarding the magnitude, contributing factors, and potential solutions to road traffic collisions, injuries, and deaths. We designed a qualitative study of 24 in-depth interviews with 14 vulnerable road users (pedestrians, occupants of powered 2- and 3-wheelers, cyclists) and ten non-vulnerable road users in four high-risk areas in November 2022. We used a mixed deductive (direct content analysis) and inductive (interpretive phenomenological analysis) approach. In the direct content analysis, a priori categories based on Haddon's Matrix covered human, vehicle, socioeconomic environment, and physical environment factors influencing road traffic collisions, along with corresponding solutions. We used inductive analysis to identify emerging themes. Participants described frequent and distressing experiences with collisions, and most often reported contributing factors, implementation gaps, and potential solutions within the human (road user) level domain of Haddon's Matrix. Implementation challenges included sporadic enforcement, reliance on road users' adherence to safety laws, and the low quality of the existing infrastructure. Participants expressed that they felt neglected and ignored by road safety decision-makers. This research emphasizes the need for community input for successful road safety policies in Ghana and other low- and middle-income countries, calling for greater governmental support an action to address this public health crisis. We recommend the government collaborates with communities to adapt existing interventions including speed

**Data Availability Statement:** All relevant data are within the manuscript and its Supporting Information files.

**Funding:** Research reported in this publication was supported by the Fogarty International Center of the National Institutes of Health under grant #D43TW009345 awarded to the Northern Pacific Global Health Fellows Program and R35-TW009345. These funded the work of Dr. Aldina Mesic, Dr. Charles Mock, Dr. Adam Gyedu, and Dr. Barclay Stewart. The content is solely the responsibility of the authors and does not necessarily represent the official views of the National Institutes of Health. The funders had no role in study design, data collection and analysis, decision to publish, or preparation of the manuscript.

**Competing interests:** The authors have declared that no competing interests exist.

calming, footbridges, and police enforcement, and introduces new measures that meet local needs.

## Introduction

Road traffic collisions (RTCs) are one of the leading causes of death globally, resulting in 1.35 million annual deaths [1]. The vast majority (90%) of deaths occur in low- and middle-income countries (LMICs). Although there is a staggering burden of RTCs, injuries, and deaths in LMICs [1, 2], community participation in injury prevention is insufficient [1, 3, 4], potentially due to limited resources for injury prevention, and top-down approaches to decision-making [4–6]. Community engagement plays a pivotal role in global health and is widely believed to positively impact health and reduce inequalities [7–9]. Participation is regarded as both a means to achieve the objectives of health programs efficiently and effectively and as an end in itself, empowering communities to take control of their health and development [8, 10]. Since 1987, the World Health Organization (WHO) has emphasized that community participation is a novel and important approach to reduce road traffic injuries., stating national-level road safety decisions are more likely to be efficiently implemented with the backing of communities [7, 11]. Participation may help ensure that interventions address the population's needs, drawing upon local knowledge and priorities[12]. Providing communities with an opportunity to share their experiences is an important method of engagement. However, in LMICs, qualitative studies on this subject are scarce with a recent scoping review only identifying 37 such studies published between 2003 and 2017 [13]. Based on this and our literature review, we believe that only ten studies have been conducted in Africa, although it faces the greatest burden and could benefit from studies on interventions to meet population needs [14–23]. Five studies focused on contributing factors to RTCs or injuries. They reported themes relating to risky road user behavior such as drink-driving among motorbike taxis [17], drug use among commercial drivers [23], jaywalking among pedestrians [14, 19], and unsafe driving among motorists [15, 22]. Other factors included poor road conditions and traffic signs[23], narrow roadways [14], traffic congestion affecting post-crash care [20], and corruption affecting enforcement [15, 17]. Only one study focused on engagement for intervention planning by asking participants to share their views on footbridges. Participants in Dar es Salaam, Tanzania, spoke highly and positively about citizen participation and involvement in the design and plan of road safety interventions, suggesting. that footbridges could be improved by addressing the proliferation of socioeconomic activities in proximity, crowding, lack of lights at night, and the infrastructure design components, which require a certain fitness level [19].

The importance of qualitative research in this context is clear. As Roberts (1997) emphasizes, "without an understanding of the lives and lay expertise of those on the receiving end of our well-meaning efforts, we risk ineffective, or worse, intrusive and harmful interventions" [24]. Quantitative studies, while valuable, might not fully capture the contextual factors that could make injuries more likely and interventions more effective. The real-world implications of this research gap are evident when we look at specific contexts, such as Ghana, a lower-middle-income country grappling with a high burden of injuries and deaths, with an estimated 24.9 road traffic fatalities per 100,000 people in stark contrast to the 8.3 per 100,000 observed in high-income countries (HICs) [1, 25]. Yet, there is a shortage of literature on road users' perspectives on road traffic injuries and deaths. While some studies have been conducted – for example, with 26 pedestrians in Krobo [14], 20 commercial drivers in Greater Accra [15], and

13 collision survivors in Accra [16]– they primarily focus on understanding contributing factors or the experiences of survivors. As before, they do not focus on road users' involvement in road safety interventions. This is worrying, given indications of both public dissatisfaction with the existing, or lack of interventions [26, 27]. Bridging this gap in literature is of growing importance considering the outcry regarding the extent and reach of road safety initiatives nationwide, coupled with the potential benefits of community participation.

This qualitative work aimed to fill this research gap by exploring the perceptions of road users, who are rarely represented in literature and decision-making, in high-risk areas along national roads on crucial issues in road safety with three guiding questions:

- What are road users' perceptions of the magnitude of RTCs, injuries, and deaths in Ghana?

- Which factors contribute to road safety issues?

- What are the current successes, gaps, and potential solutions?

By involving the community, the government can effectively implement thoughtful road safety policies that are aligned with population needs will have the highest potential for success.

## Materials and methods

### Study setting

We conducted a qualitative study with road users from November to December 2022 in Ghana. This study is part of a multi-year project with the Ghana National Road Safety Authority (NRSA), the Building and Road Research Institute (BRRI), the Kwame Nkrumah University of Science and Technology, and the University of Washington aimed at identifying spatiotemporal trends in injuries, assessing factors associated with injury severity, and conducting natural experiments on the effect of interventions. This work aims to maximize the successful implementation of interventions by engaging road users.

This study was conducted in four statistically significant crash clusters (termed hot spots) of injury severity from 2017-2020 on national roads using the nationally representative and georeferenced BRRI database (*publication forthcoming*). We purposely selected locations based on urban/rural status, traffic patterns, and recommendations from in-country partners. We selected locations to capture urban and rural views, with a focus on engaging participants in the Northern Region, which has unique traffic patterns (e.g., high numbers of powered two- and three-wheelers s) and a historically marginalized population.

### Approach

This work emerges from an interpretive perspective, which sees the world as constructed and interpreted by individuals [28]. We did not seek to understand objective truths about the causes and solutions of collisions to apply universally. Instead, we aimed to understand experiences in space and time (e.g., a high-risk area). In line with this perspective, the study was conducted in a natural setting (roadside) [28]. A mixed deductive-inductive approach guided this study, which is a commonly used approach that integrates the strengths of both methods. The participants are guided within expected areas of thematic content (the deductive component), but able to provide insights that do not fit within the framework (the inductive component) [29]. For the deductive component, the theoretical framework used was Haddon's Matrix [30], whereas an interpretive phenomenological analysis (IPA) was used for the inductive approach [31]. Haddon's Matrix, shown in *Fig* 1, has been widely applied in injury prevention and control to. categorize factors affecting collision and injury occurrence into four domains,

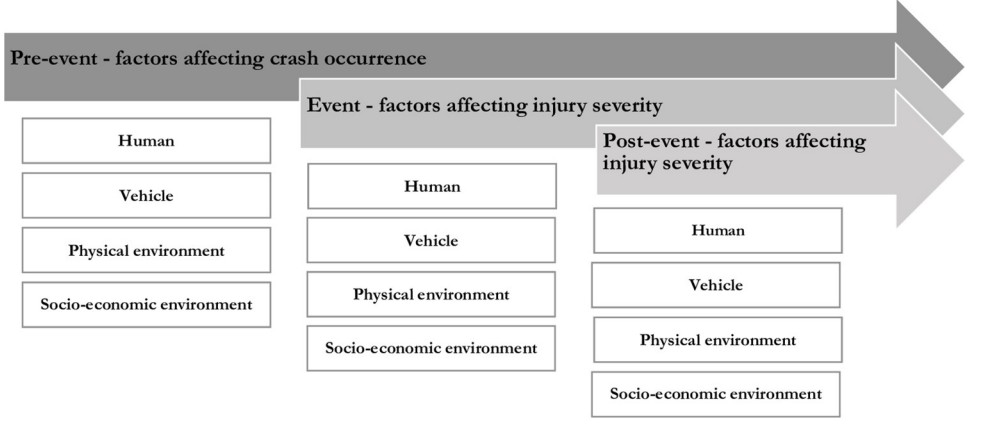

**Fig 1. Haddon's Matrix.**

including host (human factors), agent (vehicle factors), and environment (physical and socio-economic), and three phases (pre-event phase, event phase, and post-event phase, shown as arrows in *Fig 1*). We included additional questions from the Consolidated Framework from Implementation Research (CFIR) [32, 33]. We incorporated key constructs, including relative advantage, source, adaptability, and cost from the 'Innovation' domain of CFIR to examine participants' perceptions of the government's rationale for implementing road safety interventions.

Given limitations on who and what may be represented in both frameworks, we allowed for emergent themes (i.e., an inductive approach) through IPA [31]. In this approach, the data collector seeks an insider's perspective on participants' experiences as witnesses and victims of collisions and their proposed solutions in their own words.

## Study participants

We used a priori maximum variation sampling to capture a range of perspectives. The dimension of variation is road user type (vulnerable road user [VRU] versus not). VRUs are defined as those who have less collision protection than those inside a motor vehicle (e.g., pedestrians, cyclists). Non-VRUs are defined as those who do have more protection (e.g., drivers and passengers). Within each category, we ensured diversity of non-VRUs by including several types of vehicles.

Road users were approached face-to-face and defined by a screening question ("What mode of transport do you use most often?", Prompts: walking, public transport including minibuses [i.e., trotro], bicycles, motorcycles, cars, taxis, trucks). The inclusion criteria for interviews were 18 or older and consent to an interview and audio recording. Small sample sizes (9-17 interviews) are often appropriate for qualitative studies, and particularly for IPA, which involves a detailed analysis of each case [31, 34]. We aimed to recruit 16-32 participants with flexibility in sample size based on saturation.

## Data collection and analysis

We developed the semi-structured guide (included in the *S1 File)* to address our research questions within the domains/phases of Haddon's Matrix, CFIR, and in alignment with IPA principles (i.e., encouraging the person to speak with limited prompting and leading). Two Ghanaian male data collectors (SAM and BAM) conducted interviews. The data collection

team, including the two data collectors, a female field supervisor (IO), and a female lead researcher (AM) conducted and observed pilot interviews in Accra. All members had training in qualitative data collection. One data collector conducted the interview, while the other recorded and took notes roadside in a private area away from other road users. Although we proposed to recruit four to eight participants in each hot spot, we continued recruiting until we reached data saturation, defined as no new information or data (termed informational redundancy) [34]. Interviews were conducted in Twi, Dagbani, or English, depending on the participant's preference, and audio-recorded, followed by verbatim transcription/translation soon after the interview. Each transcript (defined as the interview written word for word) was quality checked by the translator and by a second team member.

Analysis was conducted in the Nvivo software [35]. We (AM and IO) conducted content analysis; the most common approach in qualitative studies on RTCs in LMICs, guided by Haddon's Framework for predetermined categories with additional inductive codes for more specific codes (e.g., excessive speeding as a factor under human-level factors) [13, 36, 37]. Coders resolved discrepancies through discussion. In IPA, each case (participant) was considered individually and reviewed for emergent themes. We report transcript numbers in the results, which are available by request. We have reported our findings using the Consolidated criteria for reporting qualitative studies (COREQ), which we have included in the *S1 File*.

### Positionality

Neither of the Ghanaian interviewers was affiliated with the government or any entities related to road safety (e.g., the NRSA, the BRRI). Therefore, we do not suspect interviewer characteristics affected how participants responded. The lead researcher, interviewers, and other authors are affiliated with the University of Washington, the Kwame Nkrumah University of Science and Technology, the BRRI, and the NRSA, which may have affected their interpretation of participants' responses.

### Ethics

This study received ethical approval from the Committee on Human Research, Publications, and Ethics at the Kwame Nkrumah University of Science and Technology (Study identification: CHRPE/AP/711/22) and the University of Washington Human Subjects Division (Study identification: STUDY00016536). We provided participants with 50 Ghanaian cedis (about 4.15 US Dollars).

## Results

### Overview of study and participants

We conducted 24 IDIs with road users in four locations: two urban hot spots in the Greater Accra Region and two rural hot spots in the Northern Region. The precise locations are shown in *Fig 2*. One urban location was at the Achimota junction intersection of National Road 1, a road spanning from the border of the Republic of Côte d'Ivoire to the border of Togo, and National Road 6 from Accra to Kumasi. Both rural hot spots were located on National Road 10, beginning in Kumasi, and ending at the Burkina Faso border, south of Tamale, 54 km and 86 km from the city's center, respectively.

We present the characteristics of the 24 participants in *Table 1*. The mean age was 34 years. Most were male. We had 14 primarily VRUs (i.e., those who said they walked, cycled, or used a powered two- or three-wheeler to travel most often) and 10 non-VRUs (i.e., those who said

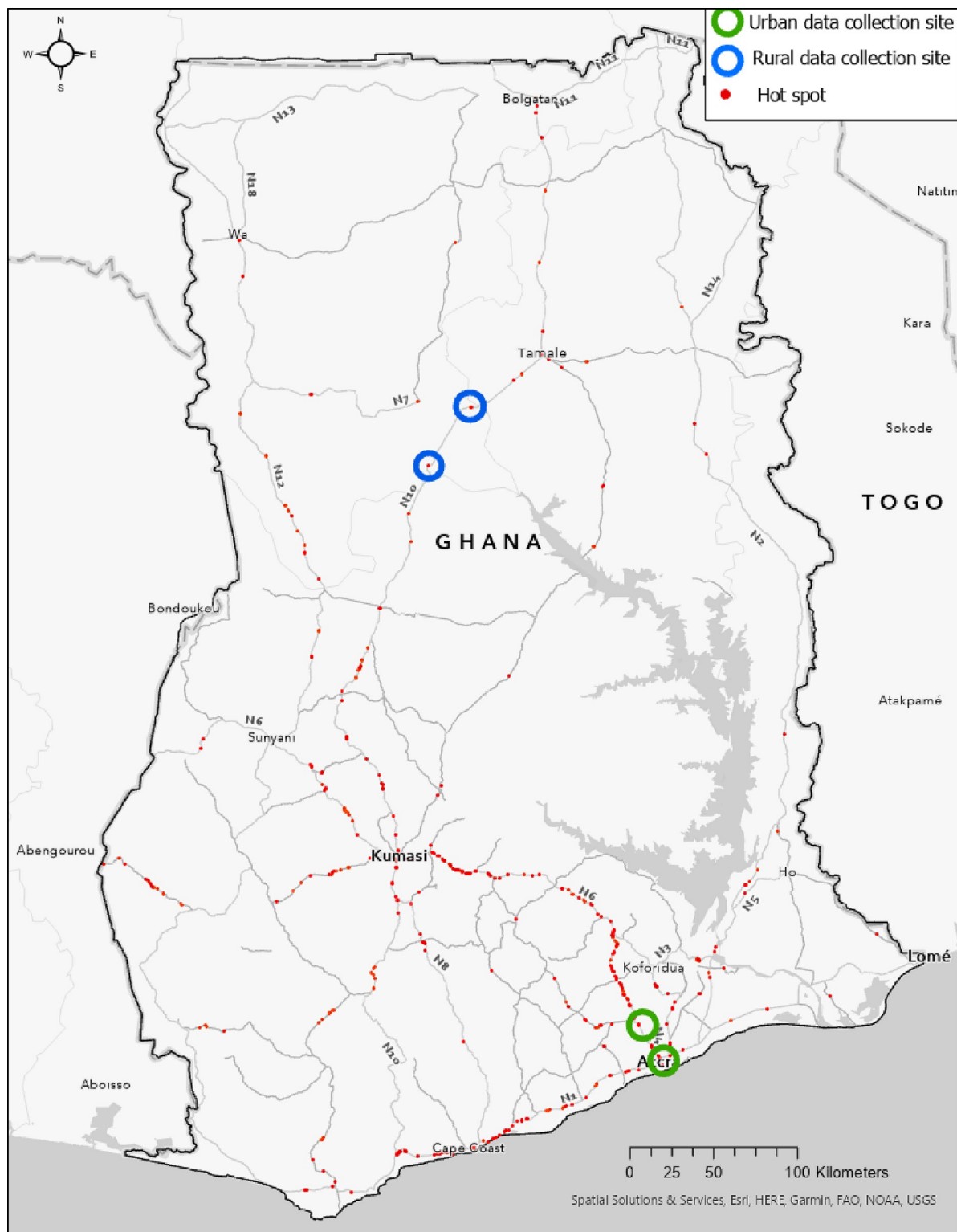

**Fig 2. Data collection sites in hot spots of RTCs, injuries, and deaths.**

**Table 1. Participant characteristics (N = 24).**

| Characteristic | Total, n (%) |
|---|---|
| Location | |
| Urban hot spot | 12 (50%) |
| Rural hot spot | 12 (50%) |
| Age, mean (SD) | 34.04 (9.4) |
| Sex | |
| Male | 21 (87.5) |
| Female | 3 (12.5) |
| Language | |
| English | 6 (25%) |
| Twi | 14 (58.3%) |
| Dagbani | 4 (16.7%) |
| Primary vulnerable/non-vulnerable status | |
| Vulnerable | 14 (58.3%) |
| Non-vulnerable | 10 (41.7%) |
| Primary road user type | |
| Driver | 17 (70.8%) |
| Passenger | 3 (12.5%) |
| Pedestrian | 4 (16.7%) |
| Primary vehicle type[A] | |
| Car | 8 (33.3%) |
| Minibus | 2 (8.3%) |
| Powered two-wheeler | 7 (21.2%) |
| Powered three-wheeler | 3 (12.5%) |
| No vehicle (i.e., pedestrian) | 4 (16.7%) |

A. These variables were developed from the response: "How you usually get around? How do you get to work?" to understand participants' primary means of transport

they use a vehicle or bus to travel most often). Interviews lasted an average of 25 minutes (range: 13-60 minutes).

## The impact of road traffic injuries and deaths

Most participants believed RTCs, injuries, and deaths were major issues in the hot spot. All agreed on the magnitude of the problem nationally, like a *46-year-old-man*, *car driver*, *rural*, *transcript 118* who commented that "accidents are rampant." Participants described how often they worried about the risk of RTCs and how it affected their behavior. They stated road safety issues affected their daily actions with examples such as people avoiding certain areas due to the risk and substantial emotional toll, saying: "If you witnessed an accident here [in the hot spot], I tell you if you are not emotionally strong, you won't want to come here again. You won't be able to walk here." – *26-year-old man*, *motorcyclist*, *rural*, *transcript 115*. Other participants described the difficult experience of traveling or having loved ones travel using the terms "fear," "panic," "scary," "horrible," "terrible," and "unbearable.". Participants reported calling on religious beliefs to arrive safely at their destination.

"Today in Ghana, there is fear and panic, especially if a relative tells you that I am traveling. The entire family begins to put the destiny of that relative in the hands of God for fear of

not returning to them. That is the extent to which accidents are affecting the country." – *25-year-old woman, pedestrian, urban, transcript 108.*

Nearly all participants reported distressing experiences of witnessing or being involved in a crash. The four participants involved in RTCs were vehicle passengers or drivers, and two were injured. Over half of the participants reported witnessing a child-involved crash, most often when a child was crossing the road. Participants emphasized the negative impact of injuries and deaths, particularly when children are harmed, which also affects national economic development, stating: "The child is a future somebody, but as he comes from school, a car will come and knock him down. Don't you see all these things send us back [in terms of development]?" – *25-year-old man, motorcyclist, rural, transcript 122.* Another participant saw this issue through the lens of missed future opportunities: "You don't know who will grow to become an important person and who is going to help Ghana in the future." – *26-year-old man, motorcyclist, rural, transcript 115.*

## Factors contributing to RTCs, injuries, and deaths

Participants had many potential explanations for the immense burden of RTCs, injuries, and deaths, which we have displayed within domains of Haddon's Matrix in *Fig 3*. We summarize key factors below.

On human-level factors, participants called out specific characteristics (e.g., being a roadside seller, termed hawker) and behaviors (e.g., excessive speeding or crossing a road carelessly) that contribute to higher risks. Vehicle factors included the poor condition, the lack of safety features, and overcrowding/overloading. Physical environment factors included poor road conditions, mixed traffic, limited speed calming measures, and disabled and unattended vehicles on the roads or shoulders. Socioeconomic and environmental factors included issues

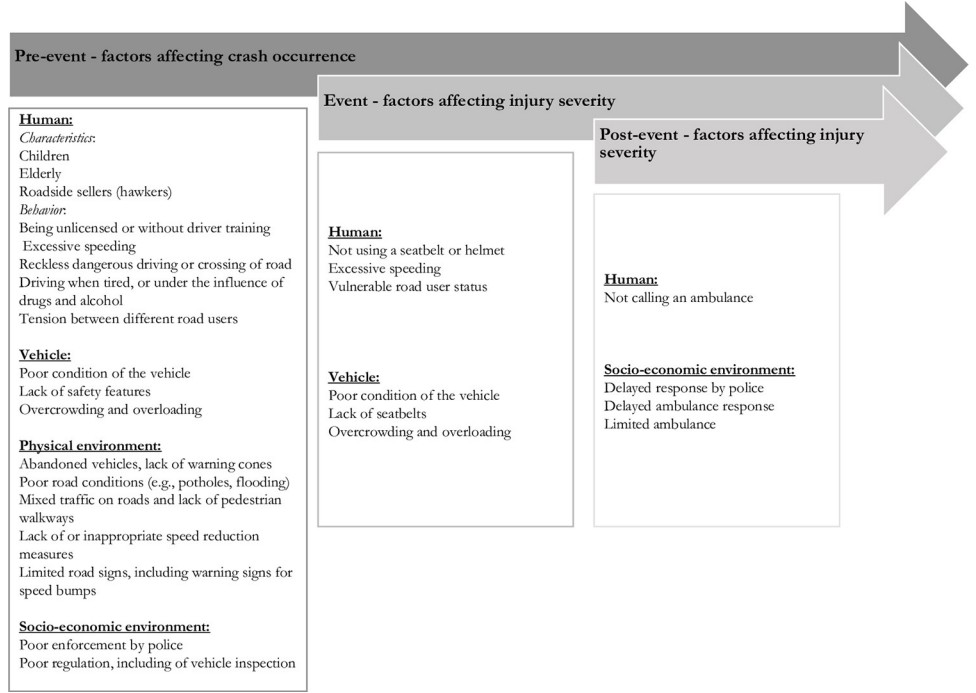

**Fig 3. Contributing factors within the Haddon's Matrix according to IDI responses.**

related to the police such as limited, questionable, or low-quality enforcement, and delayed ambulance response times after an RTC. Two notable physical environment findings were that disabled and unattended vehicles and a lack of signs or warning before a speed bump(s) could cause RTCs, injuries, or damage to vehicles. Most participants stated these features (disabled cars and speed bumps) could be challenging to see, particularly at night given limited street light infrastructure. In addition, they stated that disabled vehicles are often on the road for days or weeks. Although rarely stated as a definitive cause of RTCs, injuries, and deaths, one notable recurring emerging theme among participants was the association of various factors were with socioeconomic conditions. First, participants described low economic status underpinning the increased risk of adverse outcomes for specific people, such as hawkers that must sell goods near cars: "Someone selling on the pavement, it's not right for people to do that because it was meant for people to pass, but because of the system and economic hardship, people are here to trade." – *46-year-old man*, *minibus passenger*, *urban*, *transcript 102* and illegal motorcycle taxis (termed okadas): "I will make the okada riders to stop work for I have seen that its due to economic hardship that is making them do that."—*21-year-old man*, *commercial driver*, *urban*, *transcript 110*. Second, participants mentioned that the observed vehicle issues, including tire, brake, and safety concerns, arise from drivers being unable to afford repairs or replacements.: "It [the car] does not deserve to be on the road at all. But because of economic hardship, he's still working with it." – *46-year-old man*, *minibus passenger*, *urban*, *transcript 102*. Private and commercial drivers were no exception: "Some buses carrying passengers[. . .] will say 'let me go and manage if I get money, I will change the tire.'" – *26-year-old man*, *motorcyclist*, *rural*, *transcript 115*.

## Current road safety successes and gaps

Most participants were aware of existing measures, including speed bumps, pedestrian footbridges, and educational campaigns. When prompted on the government's road safety efforts, opinions differed. Most acknowledged the existing efforts but emphasized the government's need to address the burden of RTCs, as noted by a *39-year-old man*, *powered three-wheeler driver*, *rural*, *transcript 116*, stating, "we are pleading to the government to do more. What they are doing is not enough." Participants often referred to elected officials as the government, with a *48-year-old man*, *car driver*, *urban*, *transcript 109* stating that they only "care[. . .] about their pocket and their position." Some believed policymakers do not keep promises they make during elections, stating: "They will just say it, but when they get the power, they don't do it." – *46-year-old man*, *minibus passenger*, *urban*, *transcript 102*. Participants described officials spending substantial time discussing rather than acting on the problems. Of note, participants spoke broadly about the government or elected officials, rather than calling out specific agencies.

Despite negative comments, participants described successes including improved road conditions, repaired potholes, new/resurfaced roads, reflective paint, and pedestrian footbridges. Regarding footbridges, participants described the positive impact in the hotspot: "before the footbridge, cars used to knock people every day, in and day out. But after the construction of the footbridge, now it has reduced." – *46-year-old-man*, *minibus passenger*, *urban*, *transcript 102*. Other successes included police, with one participant reporting "the police do their best to protect us." – *35-year-old man*, *powered three-wheeler driver*, *urban*, *transcript 107*, and the proliferation of ambulances: "We just got a new ambulance, [. . .] anytime we call them they come fast."– *28-year-old man*, *motorcyclist*, *rural*, *transcript 121*.

Participants described the equitable response of ambulances. When prompted on any disparities in access (e.g., urban versus rural, ability to pay), they stated that ambulances would

arrive regardless of any such factors: "No, they don't look at any person, [. . .] whom you are, where you are coming from. They do help if it is anyone at all." – *26-year-old man, motorcyclist, transcript 124*. However, others stated that ambulances take a long time to arrive: "The last time they[bystanders] called an ambulance, [they said] it would be about an hour." -– *32-year-old man, car driver, rural, transcript 123*. Additionally, it was reported that individuals rarely call ambulances after a collision and instead rely on community members and Good Samaritans for transport to the hospital.

Participants described implementation gaps of existing interventions, citing issues such as the absence of interventions overall or at specific times and locations: "During the day, they [the police] enforce the law very well [. . .]. But in the night, where they are gone is where most of the accidents occur." – *48-year-old man, car driver, urban, transcript 109*. Beyond gaps, they also acknowledged limitations of existing interventions, such as speed bumps being poorly constructed: "Some of the speed bumps don't last. They don't include enough [. . .] cement." – *41-year-old man, car driver, rural, transcript 120*, and road users not utilizing existing infrastructure designed to increase safety, as a *46-year-old man, minibus passenger, urban, transcript 102* highlights by concluding "due to our stubbornness and laziness, we don't use the footbridge. We prefer to cross the road instead." Views on enforcement were mixed, but some participants commented on police corruption: "instead of the police arresting the driver, they will take a token from them and allow them scot-free" – *a 46-year-old man, minibus rider, urban, transcript 102*, which may undermine road safety improvement efforts.

### Engagement in road safety

Most participants stated they had no experiences to indicate their views are considered in road safety decision-making. A *28-year-old man, motorcyclist, rural, transcript 121* stated "the government doesn't come and ask about the thing we want," leading to reports of feelings of neglect among other participants: "They don't respect our lives." – *35-year-old man, motorcyclist, rural, transcript 122*. Some participants described taking road safety matters into their own hands. For example, a *41-year-old man, car driver, transcript 120* reported, "[in] this place, accidents were [. . .] too much, we decided to make our own speed rump," as the government was not responding to their requests promptly.

In contrast, a few participants described positive engagement and empowerment. For example, a *34-year-old man, motorcyclist, rural, transcript 122* eagerly noted, "when people complained to the district assembly, they came and did the speed ramp for us." Another respondent was pleased to be interviewed by a news station after a severe RTC occurred:

> "There was an accident here [. . .] which caused a very long traffic [so] that people were not able to go to work. Some reporters from [. . .] the television station came around to interview us, and I really poured my heart out to them. I explained to them the reasons for frequent accidents in the area, and as you can see now, the road is being expanded to help reduce accidents. This actually proves to me that the government really listens to us." – *30-year-old woman, pedestrian, urban, transcript 112*.

### Proposed road safety solutions

We offered participants an opportunity to suggest measures for reducing RTCs, injuries, and deaths. *Table 2* presents the proposed solutions and the related contributing factors within Haddon's Framework.

**Table 2. Contributing factors and corresponding proposed solution.**

| Haddon's Matrix Level | Proposed solution | Illustrative quotes |
|---|---|---|
| Human | Contributing factor: drivers driving under the influence of drugs, alcohol, or when tired | |
| | Drivers, and particularly commercial drivers, should reduce drinking, drug, driving while tired | "What I think will decrease the risk of accidents here [is that] the buses, they should give them two drivers [. . .]. [With] one driver, when he is tired, it can cause an accident [. . .]. When there are two drivers, if one feels like sleeping, then one will also take over." – *41-year-old man, car driver, rural, transcript 120.* |
| | Contributing factor: road users not obeying or knowing the road safety laws | |
| | Education for road users, including pedestrians, children, and the general public<br>Training for drivers on how to safely drive | "Training – the government has to bring it onboard for them[drivers] to learn before they get on the road." *46-year-old man, minibus passenger, urban, transcript 102.*<br>"I will add road safety laws to our educational curriculum so that it will enlighten their [children's] knowledge." – *35-year-old man, powered three-wheeler driver, urban, transcript 107.*<br>"Educate the general public on road safety." – *56-year-old man, car driver, urban, transcript 105.* |
| | Contributing factor: drivers drive carelessly, recklessly, and speed excessively | |
| | Drivers should reduce careless, reckless driving, and pedestrians need to reduce dangerous crossing of the road | "I will advise pedestrians to be careful when walking on the roadside." – *22-year-old man, minibus passenger, urban, transcript 101.*<br>"Drivers have to be careful, over-speeding and wrong overtaking." – *35-year-old man, car driver, rural, transcript 119.* |
| | Drivers should reduce excessive speeding, particularly when entering communities and busy areas | "If you [the driver] get there, slow down; people live there; some sell by the roadside." – *30-year-old woman, pedestrian, urban, transcript 112.* |
| | Contributing factor: limited helmet and seatbelt use | |
| | Drivers, passengers, and motorcyclists are urged to always wear helmets and seatbelts | "The accident's impact can be reduced when you are wearing a helmet."– *28-year-old man, motorcyclist, rural, transcript 121.*<br>"We will plead with drivers to wear their seatbelt." – *30-year-old woman, pedestrian, urban, transcript 112.* |
| Vehicle | Contributing factor: vehicles are not fitted with seatbelts, enough seats, or safety features | |
| | Seatbelts should be mandated, installed in vehicles | "Every vehicle is supposed to have seat belts and enough space." – *28-year-old man, motorcyclist, rural, transcript 121.*<br>"For cars, if you are onboarding, your safety should be guaranteed. If there is an accident, since all those things that protect the passengers are not available, the accident will be fatal."– *26-year-old man, car driver, urban, transcript 106.* |
| | Cars, buses, and powered three-wheelers should not be overloaded with passengers or goods | "When they [vehicle owners] are fixing the seat, they should position the seat with intervals." – *30-year-old woman, pedestrian, urban, transcript 112.* |
| | Contributing factor: poor condition of vehicles | |
| | Vehicles should be inspected for roadworthiness or removed from roads | "Some of the vehicles [. . .] need to be seized." – *28-year-old man, motorcyclist, urban, transcript 111.* |
| Physical environment | Contributing factors: poor road conditions, limited signs, including warning signs for speed limits | |
| | Improve roads, reduce curves, improve signs, including for speed bumps | "If they were to be a listening government, Kumasi to Accra Road should [. . .] be made dual." – *48-year-old man, car driver, urban, transcript 109.*<br>"They should reconstruct the gutter here so that any time it rains, there will be no flood."– *21-year-old man, commercial driver, urban, transcript 110.*<br>"Put up road signs to guide new or foreign drivers to their destination." – *30-year-old woman, pedestrian, urban, transcript 112.*<br>"They should construct the speed bump with reflected stripes to alert the drivers from a distance. – *35-year-old man, car driver, rural, transcript 119.* |
| | Contributing factor: numerous disabled vehicles on or alongside the road, limited warning triangles to tell drivers about the upcoming disabled vehicles | |
| | Reduced disabled vehicles on the road, put warning signs before disabled vehicles, and remove warning signs after the vehicle is removed | "The people around should help push the car to the proper place, or the police should get a towing car to tow the car off the road [. . .] 24/7 every day." – *26-year-old man, car driver, urban, transcript 106/* |
| | Contributing factors: mixed traffic on roads and lack of pedestrian walkways | |
| | Separate road users (including for trade), build infrastructure such as sidewalks and footbridges | "I will construct separate roads for bicycles, motorbikes, and cars." – *22-year-old man, minibus passenger, urban, transcript 101.*<br>"I would move all the hawkers by the roadside to a very safe place allocated to them." – *35-year-old man, tri-cycle driver, urban, transcript 107.* |
| | Contributing factor: limited speed calming measures | |
| | Install speed measures, including speed bumps, speed cameras, and speed governors (in-vehicle speed monitoring) | "More speed bumps should be constructed." – *22-year-old man, minibus passenger, urban, transcript 101.*<br>"I will put a machine inside the car to limit their speed; if you speed beyond that limit, it will not go. I will also fix cameras in town to check [the drivers'] speed limit." – *56-year-old-man, car driver, urban, transcript 105.* |

*(Continued)*

**Table 2.** (Continued)

| Haddon's Matrix Level | Proposed solution | Illustrative quotes |
|---|---|---|
| Socioeconomic environment | Contributing factor: limited ambulances and delayed ambulance response | |
| | Educate road users on first aid and on the transport of injured | "People selling don't have any knowledge about first aid, so the government should select people to come and educate them." – *46-year-old man, minibus passenger, urban, transcript 102.* |
| | Increase or strategically allocate ambulance and health centers | "I will increase the number of ambulances." – *28-year-old man, motorcyclist, rural, transcript 121.*<br>"Since this place is busy, if we get an ambulance as a standby, it will help." – *56-year-old man, car driver, urban, transcript 105.* |
| | Contributing factor: poor or limited enforcement | |
| | Increase enforcement, including with military officials | "Where there is a footbridge, […] I will employ the service of the military men and policemen […] to enforce the road safety laws."– *35-year-old man, powered three-wheeler driver, urban, transcript 107.* |

## Discussion

The gravity of road safety issues in Ghana and other LMICs is substantial, with participants in this study demonstrating a thorough understanding of the problem's magnitude. Participants described their frequent and distressing experiences of crashes, detailing the impact on their behavior. In contrast, policymakers have yet to fully grasp its significance. While the NRSA saw a notable increase in annual funding with $5.06 million in 2020/2021, compared to 3.76 million in 2018/2019, this progress is inadequate to address this crisis [38]. The economic toll inflicted by road-related severe injuries and fatalities ($4,507,000,000 estimated losses, 8.2% of GDP in 2016), emphasizes the urgency of investment [25]. Despite the 1.3 million deaths, 50 million injuries, and macroeconomic loss estimated at $1.8 trillion from 2015-2030 globally, this issue is often sidelined to other health conditions. The escalating trends of urbanization and motorization forecast a worsening scenario without urgent action. Therefore, we implore leaders in Ghana and other LMICs to acknowledge this crisis and allocate resources accordingly. To put it simply: "we are pleading for the government to do more."

In line with this plea, we highlight participants' perceptions on the contributing factors, gaps, and potential solutions. Respondents noted several key contributors to crash frequency and severity, complementing existing literature [1]. Participants ubiquitously reported human-level (road-user) factors, implementation gaps, and proposed solutions, such as risky road user behavior and the need for increased police enforcement. This narrow view may reflect road users' familiarity with existing policies, which largely emphasize prosecuting drivers and public education [39]. Previous studies corroborate these themes, stressing road users' personal responsibility for their own safety [17] and attributing injuries and deaths to risky behaviors like drug use and speeding [14, 15, 21–23]. These findings also align with the global focus on changing road user behavior [40]. While road users undoubtedly share a collective responsibility for safety, adapting the road network to accommodate human vulnerability is essential, per the leading Safe Systems Approach [41, 42]. In this framework, blaming road users is inappropriate if the system fails to address these vulnerabilities. Further, behavior change strategies can face implementation challenges and may have limited effectiveness [40]. A recent review of road traffic injury prevention in LMICs observed mixed results for road user-focused efforts, including public awareness and educational campaigns [3, 43]. Increased police presence led to a reduction in deaths in Kampala [44], while drunk-driving enforcement in Mexico led to fewer crashes but did not affect injuries or fatalities [45]. Participants often described the need for speed management interventions as a road user-level intervention, such as automated speed enforcement cameras and speed governors, which have been implemented

nationally in Rwanda [46, 47]. Automated speed enforcement is currently being piloted in Ghana, indicating a program and population need alignment.

Participants identified the poor condition of vehicles as a key factor and potential solution, aligning with prior research linking non-genuine car parts, substandard mechanic work, and inadequate repair supervision to RTCs in Ghana [48] and prior road user perception studies on the causes of RTCs in Africa [20, 22]. The absence of safety features was stated to increase the likelihood of occupant injury or death. Given that Ghana does not meet some United Nations Vehicle Safety Regulations, this finding is unsurprising [25]. Within the physical environment category, respondents highlighted the high prevalence of disabled vehicles on roads and inadequate warning signs before speed calming measures. These issues mirror those raised by Ghanaian road safety experts in a recent modified Delphi study [49].

On gaps, our participants, like other Ghanaian community members, express dissatisfaction with the lack of interventions. This discontent has led to them to construct unauthorized speed-calming measures, as has been documented in news articles nationally [50–52]. Speed bumps are a highly politicized issue in Ghana, evidenced by "no speed bumps, no votes" billboards during prior elections [53]. This may indicate that policy implementation is not promptly meeting Ghanaians' needs, a phenomenon also documented in other African and Latin American countries [54]. The participants had similar concerns about the limitations of interventions. Gaps of existing interventions included limited enforcement, a need for road users to adhere to safety laws independently of enforcement, and subpar infrastructure interventions. For instance, participants suggested adding reflective stripes to government-constructed speed bumps and that overhead footbridges should have enforcement to ensure utilization. Such recommendations speak to the need for designing, adapting, and testing interventions and implementation strategies for the Ghanaian and LMIC context [55, 56]. Implementation issues can negatively affect acceptability, adoption, cost, penetration, and impact [55, 57]. For example, although there are positive impacts of separating road users with footbridges [40], this effect may not be observed if the footbridge is underutilized due to poor design or implementation [19, 54, 58, 59]. Existing measures, especially speed-related, are questioned for their effectiveness and suitability in prior literature. Despite studies showing speed bumps reduce injuries and deaths [26, 53, 60], excessive speeding remains common [61] due to ad hoc implementation based on varying factors rather than standardized guidelines [26, 62]. Consequently, reports of poor road safety measures and their negative effects have led to speed bump relocation or removal on major roads [27, 52] and calls for speed calming measures to be restricted to residential streets, rather than high-speed arterial roads [26, 63]. In line with this prior literature and our findings, we recommend that road safety agencies consider and document local adaptations to interventions [64]. This includes modifying the intervention itself, its delivery, evaluation, and the strategies used for implementation or spread to improve the reach, effectiveness, sustainability, and alignment with local needs [65, 66].

Proposed solutions align with global evidence-based practices including traffic calming [60, 67], improved road designs [40, 68], warning signs about hazards [69], roadside and central barrier systems [70, 71], and the separation of road users such as pedestrians [72–74], cyclists [75–77], and motorcycle occupants [78, 79]. The socioeconomic environment solutions addressed prehospital care, crucial for minimizing crash-related harm [80]. In LMICs, an estimated 54% of the 45 million deaths could be addressed with prehospital and emergency care, with unintentional injuries, including those from RTCs, contributing the most to the preventable morbidity burden [81]. Participants suggested community first aid, previously successfully with commercial drivers in Ghana [82], and strategic ambulance service allocation. However, they universally reported relying on community members and Good Samaritans for

hospital, a common scenario in LMICs [81, 83, 84], indicating potential demand-side barriers to prehospital care despite supply-focused solutions.

Despite the richness of their insights, road users felt unheard. This aligns with prior observations; out of the limited qualitative studies on African road users, only three involved Ghanaians [14–16], underscoring the importance of acknowledging this neglected population's experiences and input in the road safety dialogue. While initiatives are typically government-led (i.e., by the NRSA in Ghana), community involvement could enhance support. Current efforts include workshops, seminars, radio and TV discussions, and the operation of a road safety call center, could be extended to involve communities in problem identification and solution finding, following the WHO Safe Communities recommendations [85, 86]. This approach, particularly when targeting high-risk groups and evaluating program impacts, can promote effective interventions and foster public trust.

Our study is subject to some limitations. Despite using maximum variation sampling and achieving data saturation, the small sample size may limit the diversity of perspectives. Also, using a semi-structured questionnaire based on Haddon's Matrix could have influenced participants' responses and potentially discouraged policy- or system-level suggestions. We took a mixed deductive-inductive approach to allow for emerging themes to address this concern. There was also a risk of social desirability bias, especially concerning government or police actions, although measures were taken to minimize this (i.e., reassurance that lead researchers and data collectors were not affiliated with government entities) [28, 87, 88]. Despite these limitations, we believe our study, which centers road user voices to contribute to Ghana's road safety dialogue, provides trustworthy findings.

Overall, we call for greater investment and effort into the prevention of RTCs, injuries, and deaths in Ghana and other LMICs. Researchers, policymakers, and practitioners should view this qualitative study as an example of community engagement as a valuable tool to guide impactful and sustainable interventions. By speaking directly to those affected, we developed a rich contextualized understanding of road users' perceptions on the causes, gaps, and solutions to road safety issues in Ghana. We encourage other LMICs to take a similar approach. Beyond this, we stress the need for evidence-based interventions, and for local-based adaptations of those interventions, an important lesson applicable to other settings facing a similar high burden of injuries and deaths.

## Conclusions

Guided by this qualitative study, our key recommendations are for the governments of Ghana and other LMICs' to recognize and address the pressing public health crisis of road traffic injuries and deaths. We encourage road safety agencies and practitioners to not only implement evidence-based interventions but also to ensure their relevance through local adaptations. Lastly, we urge road safety agencies to engage with communities for locally tailored solutions, as we have in this study.

## Supporting information

**S1 File. Road user semi-structured interview guide.**
(DOCX)

**S2 File. Interview transcripts.**
(ZIP)

## Acknowledgments

We are grateful to our participants for sharing their experiences and time with us.

## Author Contributions

**Conceptualization:** Aldina Mesic, Barclay Stewart, Irene Opoku, Bradley H. Wagenaar, Bilal Andoh Mohammed, Sulemana Abdul Matinue, Manal Jmaileh, Adam Gyedu, Angela Kitali, Martin Owusu Afram, Caryl Feldacker.

**Data curation:** Aldina Mesic, Irene Opoku, Bilal Andoh Mohammed, Sulemana Abdul Matinue.

**Formal analysis:** Aldina Mesic, Irene Opoku.

**Methodology:** Aldina Mesic, Barclay Stewart, Bradley H. Wagenaar, Manal Jmaileh, James Damsere-Derry, Adam Gyedu, Charles Mock, Caryl Feldacker.

**Supervision:** Charles Mock, Daniel Hardy Wuaku, Martin Owusu Afram, Caryl Feldacker.

**Validation:** Daniel Hardy Wuaku, Martin Owusu Afram, Caryl Feldacker.

**Writing – original draft:** Aldina Mesic.

**Writing – review & editing:** Aldina Mesic, Barclay Stewart, Irene Opoku, Bradley H. Wagenaar, Bilal Andoh Mohammed, Sulemana Abdul Matinue, Manal Jmaileh, James Damsere-Derry, Adam Gyedu, Charles Mock, Angela Kitali, Daniel Hardy Wuaku, Caryl Feldacker.

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
