## [Decision Letter · Decision Letter 0]

14 Nov 2023

PONE-D-23-27981“We are pleading for the government to do more”: road user perspectives on the magnitude, contributing factors, and potential solutions to road traffic injuries and deaths in Ghana.PLOS ONE

Dear Dr. Mesic,

Thank you for submitting your manuscript to PLOS ONE. After careful consideration, we feel that it has merit but does not fully meet PLOS ONE’s publication criteria as it currently stands. Therefore, we invite you to submit a revised version of the manuscript that addresses the points raised during the review process.

We look forward to receiving your revised manuscript.

Kind regards,

Fekede Asefa Kumsa, PhD

Academic Editor

PLOS ONE

“Research reported in this publication was supported by the Fogarty International Center of the National Institutes of Health under grant #D43TW009345 awarded to the Northern Pacific Global Health Fellows Program and R35-TW009345. These funded the work of Dr. Aldina Mesic, Dr. Charles Mock, Dr. Adam Gyedu, and Dr. Barclay Stewart.  The content is solely the responsibility of the authors and does not necessarily represent the official views of the National Institutes of Health.”

3. We note that Figure 2 in your submission contain [map/satellite] images which may be copyrighted. All PLOS content is published under the Creative Commons Attribution License (CC BY 4.0), which means that the manuscript, images, and Supporting Information files will be freely available online, and any third party is permitted to access, download, copy, distribute, and use these materials in any way, even commercially, with proper attribution. For these reasons, we cannot publish previously copyrighted maps or satellite images created using proprietary data, such as Google software (Google Maps, Street View, and Earth). For more information, see our copyright guidelines: http://journals.plos.org/plosone/s/licenses-and-copyright.

1. You may seek permission from the original copyright holder of Figure 2 to publish the content specifically under the CC BY 4.0 license. 

Additional Editor Comments:

The manuscript is pretty large and can be shortened. In addition, please make sure the conclusion is drawn from the results. 

Reviewers' comments:

Reviewer's Responses to Questions

**Comments to the Author**

1. Is the manuscript technically sound, and do the data support the conclusions?

Reviewer #1: Partly

Reviewer #2: Yes

Reviewer #3: Yes

Reviewer #4: Partly

2. Has the statistical analysis been performed appropriately and rigorously? 

Reviewer #1: N/A

Reviewer #2: Yes

Reviewer #3: Yes

Reviewer #4: Yes

3. Have the authors made all data underlying the findings in their manuscript fully available?

Reviewer #1: Yes

Reviewer #2: Yes

Reviewer #3: Yes

Reviewer #4: Yes

4. Is the manuscript presented in an intelligible fashion and written in standard English?

Reviewer #1: Yes

Reviewer #2: Yes

Reviewer #3: Yes

Reviewer #4: Yes

5. Review Comments to the Author

Reviewer #1: This qualitative study, conducted in Ghana, is exploring the perception of road users in high-risk areas on crucial issues in road safety. The results are undoubtedly very useful practically in Ghana; however, it is unclear what they contribute to the academic literature within the research field? The discussion emphasizes the need for urgent actions and economic investment in road safety, which is true, but could have been concluded without doing the study. I acknowledge the work done by the authors, but the narrative needs to be advanced such that further analyses are conducted to elevate the contribution scientifically, beyond the already known to new insights into the research field.

Reviewer #2: A very interesting paper. It’s very well written and discussed and requires only a few minor revisions. The results section can be shortened to make the paper easier to read. Were any of the participants given incentives and were any DPOs involved

Reviewer #3: Thank you for sharing this important work that has the potential to be seminal for grassroots advocacy, and brings a community perspective to the hidden epidemic in Ghana/pandemic in sub-Saharan Africa. This study sampled the perspectives of road users in Ghana, particularly vulnerable pedestrians, motorbike riders, and cyclists, regarding the causes and solutions for road traffic accidents and injuries. Qualitative research with 24 interviews was conducted in high-risk areas using controversial, but justifiable deductive and inductive approaches based on the Haddon Matrix. Participants highlighted human-related factors as the primary contributors to accidents, including issues with enforcement, the need for better infrastructure, and a sense of neglect by authorities. The study emphasizes the importance of community involvement in crafting effective road safety policies in Ghana and similar countries, calling for increased government support and the adaptation of existing measures while introducing new, locally tailored interventions.

This is well researched in perspective of the limitations that the authors have carefully outlined.

Minor comments/edits required

1. Kindly include a Standards for reporting qualitative research (SRQR) or Consolidated criteria for reporting qualitative research (COREQ) qualitative checklist in supplementary information (if not already sent to editorial team directly with no access granted to reviewers) to raise the standard of the submission. This can also be referenced in the methods as used in your reporting. this submission meets the criteria, and a deliberate reporting of this would be helpful.

2. I believe the authors do not intend this, but the ethics of the submission in terms of authorship positions reflects authors from the Global South being "stuck in the middle" with no first or last author positions. The Fogarty fellowship usually doubles up mentors (one from a HIC and another from the LMIC), and I would suggest that a joint last (senior) authorship is possible and would be more ideal, and more truly reflect this work if done collaboratively. The balance in number of authors is appropriate, but authors should consider a co-senior/last author statement in the acknowledgements. Contributions vary for supervision (including contextual alignment, local sponsorship, review of written work from a Southern gaze etc), and not necessarily equal senior expertise in Western epistemology.

3. Interviews were conducted in Twi, Dagbani, or English, depending on the participant’s preference. Were the interview prompts translated per standard with reverse translation? What standards were applied should be stated.

4. The authors state that they used a mixed deductive (direct content analysis) and inductive (interpretive phenomenological analysis) approach. Can this be referenced as a standard? What might be the challenge or limitations of this combined approach? Starting from the Haddon's matrix as a frame work, this would arguably be more deductive. authors should justify this with reference(s).

Other (minor) comments

5. The entire article writing style can be tightened/shortened again, starting from the introduction. This does not detract from the lessons, but a more concise presentation of the introduction and results would improve delivery of the entire paper and place emphasis where appropriate.

6. Inconsistent labeling of the participants in the text should be noted at the point of final edits. Some are out of parenthesis while others are in parenthesis.

Thank you for bringing this neglected community perspective.

Reviewer #4: Thank you for this interesting manuscript, I read it with great pleasure and I truly believe is a well-structured work.

I have some minor suggestions to improve the quality of the work

Introduction:

- While the introduction effectively outlines the community role and emphasizes the need for qualitative analysis, it lacks essential context on road injuries and road safety literature. Consider incorporating information on mobile use, attention shifting, decision making, and age-related factors in road injuries to provide a more comprehensive background.

Results, Discussion, and Conclusion:

- The results section is generally well-presented, but the link between the results and the conclusion could be more explicitly clarified. Consider providing a more detailed discussion that explicitly ties the findings back to the research question. This will help readers better understand the significance of the results and their implications.

Language and Style:

- A thorough English revision is recommended to address the tone and improve overall language clarity.

I congratulate the authors for the amazing work.

6. PLOS authors have the option to publish the peer review history of their article (what does this mean?). If published, this will include your full peer review and any attached files.

Reviewer #1: No

Reviewer #2: **Yes: **Mitchel Chatukuta

Reviewer #3: No

Reviewer #4: No

---

## [Author Response · Author response to Decision Letter 0]

29 Jan 2024

January 15, 2024

Dear PLOS One Editorial Board, 

Enclosed please find the revised manuscript, entitled “We are pleading for the government to do more”: road user perspectives on the magnitude, contributing factors, and potential solutions to road traffic injuries and deaths in Ghana.” for review. 

We appreciated the reviewers’ comments and request for minor revisions. We have addressed each comment in the marked-up manuscript (tracked edits) and in the attached table. 

Thank you for the opportunity to resubmit our manuscript. 

Sincerely, 

Aldina Mesic, MPH, PhD 

 

Reviewer 1 

Comment Response 

This qualitative study, conducted in Ghana, is exploring the perception of road users in high-risk areas on crucial issues in road safety. The results are undoubtedly very useful practically in Ghana; however, it is unclear what they contribute to the academic literature within the research field? The discussion emphasizes the need for urgent actions and economic investment in road safety, which is true, but could have been concluded without doing the study. I acknowledge the work done by the authors, but the narrative needs to be advanced such that further analyses are conducted to elevate the contribution scientifically, beyond the already known to new insights into the research field. Thank you for your time and review of this work.

Apologies if it was not clear how this work fits into the existing literature and contributes more broadly to the research field. 

Qualitative studies with communities on topics of road safety are limited. We found only ten studies in Africa and three in Ghana, which may indicate that there is a gap in understanding of the contextual factors which could make injuries more likely, or interventions effective or ineffective. We have rephrased much of our introduction to make this gap clearer.

The novelty and importance of this work not only lies in what is said, but also who is saying it. In this case, we are hearing directly from road users on their experiences and what they believe should be done to reduce the burden. 

This qualitative study does not aim to understand objective truths about the causes and solutions of road traffic collisions to apply universally, but rather aims to capture experiences and views in a time and space (road users in high-risk areas). We hesitate to generalize our findings to other contexts, as the point of qualitative studies is to provide a detailed, contextualized understanding of the experiences and perceptions of road users. 

Although we do not generalize the findings (for example, on which factors or solutions may be relevant), we do believe there are some lessons in this work for other LMICs, including those stated in the conclusion: 1) greater investment; 2) implementation and adaptation of evidence-based interventions to meet population needs; 3) community engagement. 

We have added several sentences to the discussion to make these lessons to other LMICs/the research field clearer: 

“Overall, we call for greater investment and effort into the prevention of RTCs, injuries, and deaths in Ghana and other LMICs. Researchers, policymakers, and practitioners should view this qualitative study as an example of community engagement as a valuable tool to guide impactful and sustainable interventions. By speaking directly to those affected, we developed a rich contextualized understanding of road users’ perceptions on the causes, gaps, and solutions to road safety issues in Ghana. We encourage other LMICs to take a similar approach. Beyond this, we stress the need for evidence-based interventions, and for local-based adaptations of those interventions, an important lesson applicable to other settings facing a similar high burden of injuries and deaths.”

Reviewer 2 

Comment Response 

A very interesting paper. It’s very well written and discussed and requires only a few minor revisions. The results section can be shortened to make the paper easier to read. Were any of the participants given incentives and were any DPOs involved Thank you for your thoughtful review. We have added a sentence about the incentive (50 Ghanaian cedis) for the participants’ time/participation. 

We have included details on who was included in the study in Table 1. 

We have also shortened / re-structured the discussion to improve readability. 

Reviewer 3

Comment Response 

Thank you for sharing this important work that has the potential to be seminal for grassroots advocacy, and brings a community perspective to the hidden epidemic in Ghana/pandemic in sub-Saharan Africa. This study sampled the perspectives of road users in Ghana, particularly vulnerable pedestrians, motorbike riders, and cyclists, regarding the causes and solutions for road traffic accidents and injuries. Qualitative research with 24 interviews was conducted in high-risk areas using controversial, but justifiable deductive and inductive approaches based on the Haddon Matrix. Participants highlighted human-related factors as the primary contributors to accidents, including issues with enforcement, the need for better infrastructure, and a sense of neglect by authorities. The study emphasizes the importance of community involvement in crafting effective road safety policies in Ghana and similar countries, calling for increased government support and the adaptation of existing measures while introducing new, locally tailored interventions. Thank you for your review and support of the work.

Kindly include a Standards for reporting qualitative research (SRQR) or Consolidated criteria for reporting qualitative research (COREQ) qualitative checklist in supplementary information (if not already sent to editorial team directly with no access granted to reviewers) to raise the standard of the submission. This can also be referenced in the methods as used in your reporting. this submission meets the criteria, and a deliberate reporting of this would be helpful. Thanks for this suggestion – it has greatly improved the reporting of the results. We have made updates to the manuscript in line with the COREQ items where appropriate. We have included the COREQ checklist in the supplementary materials. 

I believe the authors do not intend this, but the ethics of the submission in terms of authorship positions reflects authors from the Global South being "stuck in the middle" with no first or last author positions. The Fogarty fellowship usually doubles up mentors (one from a HIC and another from the LMIC), and I would suggest that a joint last (senior) authorship is possible and would be more ideal, and more truly reflect this work if done collaboratively. The balance in number of authors is appropriate, but authors should consider a co-senior/last author statement in the acknowledgements. Contributions vary for supervision (including contextual alignment, local sponsorship, review of written work from a Southern gaze etc), and not necessarily equal senior expertise in Western epistemology Thank you for raising this important concern. We acknowledge and agree that there are persistent issues with how local researchers are included in publications. 

As part of the Fogarty-funded Northern/Pacific Global Health Research Fellows Training Consortium fellowship, the lead author was mentored by two Ghanaian researchers and two American researchers. Each provided a valuable addition to the collaboration. The Ghanaian researchers were involved with sponsorship, ethical approval processes, data collection, and interpretation of results. The American researchers provided input on the research design and interpretation. All provided detailed feedback on the written work. Each of these mentors, Ghanaian and American, are middle authors on this publication. All authors took part in discussions about author contributions and approved of their positions before initial submission. There are also several other publications using these and related datasets in which the author order is different, reflecting a system where authors take initiative and leadership roles on different papers to provide opportunities for all to contribute over time. 

After discussions with mentors, it was determined that although the collaboration and contributions were equal across countries and institutions across the overall research portfolio, this specific publication stemmed directly from the lead author’s (Mesic) doctoral dissertation research in which she led all aspects from conceptualization through writing. Given this, it would not be appropriate to share first authorship (the student) or senior authorship (the chair of the academic committee). Other group publications that we have planned from this work will include local researchers in the first and/or senior position of authors as we rotate through interest areas, technical expertise, desire to publish, and time constraints.

Interviews were conducted in Twi, Dagbani, or English, depending on the participant’s preference. Were the interview prompts translated per standard with reverse translation? What standards were applied should be stated. Thank you for this question. The interview questions were translated into Twi and Dagbani by fluent speakers, and then quality checked by a second fluent translator. The same process was followed for the transcripts. 

The authors state that they used a mixed deductive (direct content analysis) and inductive (interpretive phenomenological analysis) approach. Can this be referenced as a standard? What might be the challenge or limitations of this combined approach? Starting from the Haddon's matrix as a frame work, this would arguably be more deductive. authors should justify this with reference(s). This is a commonly used approach and one which integrates the advantages of both approaches – guiding the respondents within expected areas of thematic (questionnaire driven) content while leaving room for the respondents to inform gaps and opportunities to learn new interpretations or directions. This hybrid approach reflects sound basis in the qualitative research arena for its benefits (Swain 2018). 

Swain J. A hybrid approach to thematic analysis in qualitative research: Using a practical example. Sage research methods. 2018 Jan 12.

We have added a sentence to the methods to clarify this. 

The entire article writing style can be tightened/shortened again, starting from the introduction. This does not detract from the lessons, but a more concise presentation of the introduction and results would improve delivery of the entire paper and place emphasis where appropriate. Thank you for this feedback. We have substantially reduced the wordiness to improve readability. 

Inconsistent labeling of the participants in the text should be noted at the point of final edits. Some are out of parenthesis while others are in parenthesis. We have adjusted the labeling for consistency. 

Thank you for bringing this neglected community perspective. Thank you for your thoughtful review. 

Reviewer 4

Thank you for this interesting manuscript, I read it with great pleasure, and I truly believe is a well-structured work.

I have some minor suggestions to improve the quality of the work Thank you. 

Introduction:

- While the introduction effectively outlines the community role and emphasizes the need for qualitative analysis, it lacks essential context on road injuries and road safety literature. Consider incorporating information on mobile use, attention shifting, decision making, and age-related factors in road injuries to provide a more comprehensive background. Thank you for this feedback. We opted not to include much information on road traffic injuries due to concerns about the length of the manuscript and to remain in scope, however, we note that information on the scale of road traffic injuries and deaths globally could improve the work.

We have added the following beginning to introduction: 

“Road traffic collisions (RTCs) are one of the leading causes of death globally, resulting in 1.35 million annual deaths (1). The vast majority (90%) of deaths occur in low- and middle-income countries (LMICs). Although there is a staggering burden of RTCs, injuries, and deaths in LMICs (1, 2), community participation in injury prevention is insufficient (1, 3, 4), potentially due to limited resources for injury prevention, and top-down approaches to decision-making (4-6). Community engagement plays a pivotal role in global health and is widely believed to positively impact health and reduce inequalities (7-9)….”

Results, Discussion, and Conclusion:

- The results section is generally well-presented, but the link between the results and the conclusion could be more explicitly clarified. Consider providing a more detailed discussion that explicitly ties the findings back to the research question. This will help readers better understand the significance of the results and their implications. We have adjusted and restructured the discussion and conclusions considering this comment. We have reduced the length, and structured it to follow the following format: 1) participants’ views on the issue overall; 2) contributing factors; 3) gaps in existing interventions; 4) proposed solutions; 5) engagement in community efforts; 6) recommendations and limitations 

Language and Style:

- A thorough English revision is recommended to address the tone and improve overall language clarity. We have reviewed the work in detail to improve clarity. 

I congratulate the authors for the amazing work. Thank you! Much appreciated.

---

## [Decision Letter · Decision Letter 1]

28 Feb 2024

“We are pleading for the government to do more”: road user perspectives on the magnitude, contributing factors, and potential solutions to road traffic injuries and deaths in Ghana.

PONE-D-23-27981R1

Dear Dr. Mesic,

We’re pleased to inform you that your manuscript has been judged scientifically suitable for publication and will be formally accepted for publication once it meets all outstanding technical requirements.

Kind regards,

Fekede Asefa Kumsa, PhD

Academic Editor

PLOS ONE

Reviewers' comments:

Reviewer's Responses to Questions

**Comments to the Author**

1. If the authors have adequately addressed your comments raised in a previous round of review and you feel that this manuscript is now acceptable for publication, you may indicate that here to bypass the “Comments to the Author” section, enter your conflict of interest statement in the “Confidential to Editor” section, and submit your "Accept" recommendation.

Reviewer #3: All comments have been addressed

2. Is the manuscript technically sound, and do the data support the conclusions?

Reviewer #3: Yes

3. Has the statistical analysis been performed appropriately and rigorously? 

Reviewer #3: Yes

4. Have the authors made all data underlying the findings in their manuscript fully available?

Reviewer #3: Yes

5. Is the manuscript presented in an intelligible fashion and written in standard English?

Reviewer #3: Yes

6. Review Comments to the Author

Reviewer #3: All my previous review comments have been adequately addressed. Your study strongly advocates for community involvement in shaping effective road safety policies in Ghana and similar countries. It calls for heightened government support, the adaptation of existing measures, and the introduction of new, locally tailored interventions to address the unique challenges identified by the study participants. I see this as a valuable contribution to the discourse on road safety, offering insights crucial for the development of targeted and community-driven interventions.

My authorship suggestions were obviously carefully considered and authors present what I would term a long-term authorship equity plan accross their collective portfolio of work. I still think that we can always do better for authors from the Global South and that this starts with careful authorship planning and assignment of responsibilities/capacity building, but I see the tension, as this is at the end of a project, and the respected authors are pointing at equity *in a portfolio*, which I hope does happen.

Thank you for giving thought to this.

This manuscript should be accepted and disseminated beyond the pages of an academic paper. Consider a policy paper/grassroots advocacy with this.

7. PLOS authors have the option to publish the peer review history of their article (what does this mean?). If published, this will include your full peer review and any attached files.

Reviewer #3: No

---

## [Editor Report · Acceptance letter]

25 Mar 2024

PONE-D-23-27981R1 

PLOS ONE

Dear Dr. Mesic, 

I'm pleased to inform you that your manuscript has been deemed suitable for publication in PLOS ONE. Congratulations! Your manuscript is now being handed over to our production team.

Kind regards, 

on behalf of

Dr. Fekede Asefa Kumsa 

Academic Editor

PLOS ONE